# Vegetation Affects the Responses of Canopy Spider Communities to Elevation Gradients on Changbai Mountain, China

**DOI:** 10.3390/insects15030154

**Published:** 2024-02-24

**Authors:** Pengfeng Wu, Lingxu Xiang, Qiang Zhao, Shuyan Cui, Abid Ali, Donghui Wu, Guo Zheng

**Affiliations:** 1College of Life Science, Shenyang Normal University, Shenyang 110034, China; xiaowu8181@126.com (P.W.); xianglingxums@163.com (L.X.); 15542190459@163.com (Q.Z.); cui.shu.yan@163.com (S.C.); abid_ento74@yahoo.com (A.A.); 2Department of Entomology, University of Agriculture, Faisalabad 38040, Pakistan; 3Key Laboratory of Wetland Ecology and Environment, Northeast Institute of Geography and Agroecology, Chinese Academy of Sciences, Changchun 130102, China; wudonghui@iga.ac.cn; 4Key Laboratory of Vegetation Ecology, Ministry of Education, Northeast Normal University, Changchun 130024, China

**Keywords:** Araneae, elevation pattern, β-diversity, guild category, community structure

## Abstract

**Simple Summary:**

Canopy spiders are important and abundant predators in canopy habitats. The responses to elevation change in the diversity and composition of canopy spiders are still largely neglected. In this study, the issue has been examined and explored. The results show that the richness of canopy spiders decreased whereas there was an increasing trend in evenness with the elevation increasing. The responses on the community composition of canopy spiders to elevation at the three taxonomic levels were different. The degree of impact of habitat factors would be reduced when raising the taxonomic level.

**Abstract:**

Forest canopies, an essential part of forest ecosystems, are among the most highly threatened terrestrial habitats. Mountains provide ideal conditions for studying the variation in community structure with elevations. Spiders are one of the most abundant predators of arthropods in terrestrial ecosystems and can have extremely important collective effects on forest ecosystems. How the diversity and composition of canopy spider communities respond to elevation changes in temperate forests remains poorly understood. In this study, we collected canopy spiders from four elevation sites (800 m, 1100 m, 1400 m, and 1700 m) on Changbai Mountain using the fogging method in August 2016. With the methods of ANOVA analysis, transformation-based redundancy analysis, and random forest analysis, we explored the responses of canopy spider communities to elevation. In total, 8826 spiders comprising 81 species were identified and the most abundant families were Thomisidae, Clubionidae, Linyphiidae, and Theridiidae (77.29% of total individuals). Species richness decreased whereas evenness increased with increasing elevation, indicating that elevation has an important impact on community structure. The pattern of absolute abundance was hump shaped with increasing elevation. We found that the community compositions at the three taxonomic levels (species, family, and guild) along the elevation gradient were obviously altered and the variation in community composition was higher at low-elevation sites than at high-elevation sites. There were 19 common species (23.46%) among the four elevations. Regression and RDA results showed that vegetation variables contributed to the variation in the diversity and composition of canopy spiders. Furthermore, the influence of factors would be weakened with the taxonomic level increasing. Therefore, our findings greatly highlight the important role of vegetation in the diversity and composition of canopy spiders and the influence is closely related to the taxonomic level.

## 1. Introduction

Understanding changes in biodiversity at latitudes and altitudes has been an important theme for centuries. Mountain ranges are known to harbor exceptionally high biodiversity [1,2] and, thus, they provide an ideal condition for studying the variation in community structure along elevational gradients [3]. Many studies have focused on the effects of elevation on plants [4], mammals [5], anurans [6], birds [7], and arthropods [8,9,10]. Furthermore, some authors have studied the influence of multiple groups from different trophic levels [11,12]. However, most of these studies were carried out on ground habitats and much less is known about the effects of forest canopy, especially on canopy arthropods.

Forest canopies play a crucial role in maintaining biodiversity and the provision of local and global ecosystem services [13,14] and 40% of the world’s terrestrial species live there [15]. Studies of forest canopies are integral to understanding biodiversity distributions, global climate change, and whole-forest interactions [16]. Canopy arthropods with several functional groups (i.e., pollinators, detritivores, and predators) are essential in forest ecosystems [17,18,19]. Spiders, sensitive to a wide range of environmental factors [20,21,22], are abundant and diverse predators in almost all terrestrial ecosystems [23]. Forest tree canopies provide spiders with shelter, sites for foraging, ovipositioning, sun-basking, sexual display, and overwintering [24]; previous studies showed spiders were also important and abundant predators in canopy habitats. For example, Basset [25] reported that spiders took the highest proportion (25% of total individuals) in the canopy arthropods in Australian rainforests. Katayama et al. [26] collected four times more spider individuals than ant individuals from the canopy in Borneo and found a negative correlation between them. However, the responses in terms of the diversity and composition of canopy spiders to elevation change are still largely neglected.

To clarify the difference in vertical distribution of the canopy spiders, an experiment was conducted on the northern slope of Changbai Mountain. Changbai Mountain is one of the 25 biodiversity hotspots in the world owing to its high species richness [27]. The environmental gradients of this mountain provide ‘natural laboratories’ to understand the influence of elevation gradient on species diversity and composition. Previous studies have examined the variation in the diversity of arthropods in relation to elevation gradients on Changbai Mountain, such as ground beetles [28,29], Cerambycidae [30], soil mesofauna [31], Oribatida [32], and Collembola [33,34,35]. The patterns in species richness were not consistent. For example, Gao et al. [30] showed a decreasing trend in the richness of Cerambycidae. Xie et al. [34] found a hump-shaped pattern in the species richness of Collembola, while Wu et al. [35] demonstrated an increased pattern in the richness of canopy Collembola with increasing elevation. The nonmonotonic abundance and richness pattern of Oribatida were exhibited by Yang [32]. Here, we hypothesize that (1) the abundance and richness of canopy spiders decrease with increasing elevation; (2) the community composition significantly changes with increasing elevation and there is more variation at low elevations than at high elevations; and (3) the diversity and composition can be well explained by vegetation properties and the influence decreases with the taxonomic level increasing.

## 2. Materials and Methods

### 2.1. Study Area

The study area is in the Changbai Mountain Nature Reserve (CMNR, 41°41′–42°51′ N, 127°43′–128°16′ E) in Jilin Province, Northeast China (Figure 1A), and is one of the world’s largest continuous widely undisturbed temperate forest ecosystems [36]. Changbai Mountain is the highest mountain in northeastern China, which breeds three rivers, namely Songhua River, Yalu River, and Tumen River [37]. Changbai Mountain lies east of Eurasia and has a typical continental temperate monsoon climate. The average annual temperature is −7 °C to 3 °C and the average annual rainfall is approximately 1400 mm to 700 mm from two meteorological stations (former: Erdao, 591 m; latter: Tianshi, 2623 m) [38]. According to Chen et al. [39] and Sang and Bai [40], five zones of vertical vegetation were identified, namely the mixed coniferous and broad-leaved forest zone (MCBF) (below 1100 m), mixed coniferous forest zone (MCF) (1100–1500 m), sub-alpine coniferous forest zone (SCF) (1500–1800 m), birch forest zone (BF) (1800–2100 m), and tundra zone (above 2100 m).

### 2.2. Experimental Design and Spider Sampling

In mid-August 2016, we investigated changes in canopy spider communities along an elevation gradient on the northern slope of the CMNR. We chose this sampling time because spiders had the highest activities. Four sites, each including four plots, were set at 800 m, 1100 m, 1400 m, and 1700 m above sea level. The distance between plots at the same site was 30 m (see more details in Wu et al. [35]). Habitats below 800 m do not belong to CMNR; thus, they are not suitable for comparisons. Five fundamental vegetation variables were recorded in each sampling plot, including tree height (TH), shrub height (SH), tree coverage (TC), shrub coverage (SC), and herb coverage (HC) (Table 1).

Canopy fogging, which is less dependent on behavioral characteristics, can provide an unbiased method for the true composition of canopy arthropods [41]. Canopy spiders were sampled by fogging with a portable thermal fogging machine (Thermal Fogger TS-35A, Shenzhen Longray Technology Co., Ltd., Shenzhen, China). In each plot, 100 funnel-like 0.5 m^2^ trays were used (sample plot area = 50 m^2^), at the bottom of which a 50 mL tube with 25 mL 95% ethanol was placed. The trays connected with ropes were placed 1.5 m above the ground (Figure 1B,C). All collected spider specimens were immediately placed in 95% ethanol and returned to the laboratory for sorting and identification. Fogging was carried out before sunrise to minimize fog-scatter, excluding days after rain, or during windy or misty conditions. Fogging was operated for 20 min and 2 L of a 2.2% solution of pyrethroid dissolved in diesel oil was used in each plot. To prevent erroneous sampling of arthropods from the lower vegetation layer, small trees were bent and tied to the ground or shaken to remove spiders before fogging (see more details in Zheng et al. [42]).

### 2.3. Spider Identification and Statistical Analyses

All spiders were examined and identified to a family and then to a species/morphospecies level. Some juveniles that could not be identified were not included in the data for further analysis. Hence, spiders (family level) were grouped into four guilds based on their web-building and prey-catching behaviors, according to Sørensen [43] and Floren et al. [44]: ambush predators (AP), cursorial hunters (CH), orb weavers (OW), and sheet-line weavers (SLW) (Appendix A). The Hill number of species richness (^0^*D*, q = 0), exponential Shannon diversity (^1^*D*, q = 1), inverse Simpson diversity (^2^*D*, q = 2), and inverse Berger–Parker (^3^*D*, q = 3) were calculated according to Chao et al. [45]. In addition, the Pielou evenness index was used to represent species evenness [46]. All specimens were deposited in the Northeast Institute of Geography and Agroecology, Chinese Academy of Sciences, Changchun, China.

We regarded vegetation properties, community abundance, and diversity (Hill numbers and evenness) as response variables, to evaluate the changes with elevation alteration using ANOVA followed by the least significant difference (LSD). Prior to ANOVA, the data for response variables were subjected to Shapiro–Wilk and Bartlett tests to check for normality and homogeneity of variance, respectively. When necessary, the data were transformed by natural logarithms to improve normality. The community overlap and differences were determined to illustrate the number of shared and non-shared species using a Venn diagram from the VennDiagram package [47]. In addition, differences in community composition among elevations were assessed using the Bray–Curtis dissimilarity index (β-diversity). The pattern was illustrated using the non-metric multidimensional scaling (NMDS) method followed by an analysis of similarities (ANOSIM, permutations = 999) [48].

Principal component analysis (PCA) was performed to extract vegetation information. The first axes of PCA of vegetation (PC1), accounting for 55.9% of the variation in the five vegetation variables above, was used as a predictor [49]. Then, linear regressions were used to estimate the effects of vegetation on spider diversity. Transformation-based redundancy analysis (tb-RDA) with Hellinger transformation was run to visualize the effects of habitat properties (elevation and vegetation variables) on spider composition. To obtain the predictors of variation in the diversity of canopy spiders, a random forest analysis was conducted and the relative importance (increase in mean square error percentage) was estimated using the package randomForest [50]. A permutation multivariate analysis of variance was performed to test for differences in overall models. Following He et al. [51], the significance of each influencing variable was also assessed with the “envfit” function (permutations = 999) in the “vegan” package. The main packages involved in the analyses were vegan [52] and ggplot2 [53]. All statistical analyses were conducted in R version 4.2.1 [54].

## 3. Results

Across the elevation gradient, 8826 spiders were identified, representing 15 families and 81 species (Appendix A). Linyphiidae had the highest richness, representing 30.87% of the total species, followed by Araneidae (17.29%), Thomisidae (14.82%), Theridiidae (8.64%), and Salticidae (6.18%). The most abundant family was Thomisidae (28.28%), followed by Clubionidae (22.94%), Linyphiidae (15.59%), and Theridiidae (10.48%). Combined with three other families (Araneidae, Tetragnathidae, and Salticidae), they comprised the main body of the canopy spiders on Changbai Mountain. The other eight families merely made up 1.16% of the total individuals.

### 3.1. Response of Canopy Spider Diversity to Changes in Elevation

The value of species richness (^0^*D*) was significantly higher at 800 m and 1100 m than that at 1400 m and 1700 m (Figure 2A, *p* < 0.001), while it was very similar between the two lower elevations and between the two higher elevations. The highest value of exponential Shannon diversity (^1^*D*) was obtained at 800 m, followed by 1400 m, 1100 m, and the lowest at 1700 m with a marginal significance (Figure 2B, *p* = 0.058). The Pielou evenness index was significantly higher at 1400 m than at 1100 m (Figure 2E, *p* = 0.041), while no significant differences were exhibited between other elevations. In addition, the inverse Simpson diversity (^2^*D*) and inverse Berger–Parker (^3^*D*) had no significant differences among the four elevations (Figure 2C, *p* = 0.155; Figure 2D, *p* = 0.196).

### 3.2. Response of Community Abundance and Composition of Canopy Spiders to Changes in Elevation

Total absolute abundance showed a hump-shaped pattern among the four elevations, peaking at 1100 m (Appendix A, *p* > 0.05). The top four species ranked were *Clubiona mandschurica* (1928 individuals, 22%), *Xysticus emertoni* (1439 individuals, 16%), *Lysiteles silvanus* (631 individuals, 7%), and *Drapetisca socialis* (364 individuals, 4%). Except the first species, the relative abundance of the other three species was extremely significant among elevations (Figure 3A, *p* < 0.01). The relative abundance of *X. emertoni* was significantly greater at 1100 m and 1700 m than that at 800 m. *L. silvanus* and *D. socialis* both showed hump-shaped patterns, which were higher at 1100 m and 1400 m, respectively. At the family level, Thomisidae, Clubionidae, and Linyphiidae were of dominance at every assessed elevation (Figure 3B, Appendix A). The relative abundance of Thomisidae peaked at 1100 m, followed by a peak at 1700 m, which was significantly higher than that at 800 m and 1400 m. Linyphiidae and Theridiidae had the highest values at 1400 m with both being of significance among elevations, while the value of Araneidae was significantly higher at 1700 m (Figure 3B and Appendix A, *p* < 0.05). Clubionidae and Tetragnathidae exhibited no significant variations among elevations (Figure 3B and Appendix A, *p* > 0.05). Interestingly, Salticidae was the dominant family at the elevation of 800 m (17.80%), whereas it drastically decreased at 1100 m (3.31%) and was not found at higher elevations (Appendix A, *p* = 0.003). At the guild level, there were extremely significant differences in the relative abundance for all guilds among the four elevations (Figure 3C, *p* < 0.01). The relative abundance of CH showed a monotonically decreasing trend, while that of AP, SLW, and OW had the highest value at 1100 m, 1400 m, and 1700 m, respectively.

The Bray–Curtis dissimilarity (β-diversity) showed marginal significance within the four elevations (Figure 4A, *p* = 0.058). β-diversity at 800 m was the highest, indicating a large difference in composition among sites at the lowest elevation. Furthermore, NMDS ordination plots revealed significant spatial segregation of spider communities among elevations except slightly overlapping at 1400 m and 1700 m (Figure 4B, stress = 0.0402, *p* = 0.001).

### 3.3. Effects of Habitat Factors on the Canopy Spider Community

Absolute abundance, species richness (^0^*D*), and exponential Shannon diversity (^1^*D*) of canopy spiders negatively correlated with the vegetation variable (Figure 5), with significance (Figure 5B, *p* < 0.001) and marginal significance (Figure 5C, *p* = 0.051) in ^0^*D* and ^1^*D*, respectively. Random forest analysis explained 46.9% of the variation in richness and demonstrated that TH, TC, and HC were the important predictors (Figure 5D). The RDA results illustrated that habitat variables impacted the canopy spider composition in a varying degree at different taxonomic levels. All habitat variables significantly affected canopy spider communities at the species level (Figure 6A; first canonical axis, *R*^2^ = 38.3%, *p* = 0.001; all canonical axes: *R*^2^ = 54.3%, *p* = 0.001) and at the family level (Figure 6B; first canonical axis, *R*^2^ = 22.6%, *p* = 0.004; all canonical axes: *R*^2^ = 32.9%, *p* = 0.02) except SC. However, only elevation and SH significantly affected canopy spiders at the guild level (Figure 6C; all canonical axes: *p* = 0.376), with the variance explained being 5.8%. Elevation was the most important factor at the three levels above, yet its importance decreased quickly with the taxonomic level increasing. TC, HC, and SH were the second most important variable at the species, family, and guild level, respectively (based on *R*^2^ and *p*-value). The effect of HC was opposite to that of the other four vegetation variables (TH, SH, TC, and SC), which were closely related to the site at low elevation (800 m).

## 4. Discussion

Forest canopies are among the most highly threatened terrestrial habitats globally [42]. To the best of our knowledge, canopy studies are comparatively few and most of them have been conducted in the tropical regions [26,42,43,55]. In this study, we selected a typical temperate area (CMNR) as our research site and explored the effects of elevation on the diversity and composition of canopy spiders and the mechanisms underlying these changes. This study represents a temporally-constrained ‘snapshot’ of the canopy spiders and the results should be interpreted with limitations. For instance, certain species may exhibit activity peaks during other seasons.

### 4.1. Difference in Diversity of Canopy Spiders

Generally, there are two major elevation patterns in richness, namely hump-shaped and monotonically decreased [56,57,58,59]. Partly supporting our hypothesis 1, the species richness of canopy spiders along the elevation gradient on Changbai Mountain followed a monotonically deceased pattern, which has been well documented for arthropods [60,61,62]. For example, Burwell and Nakamura [60] found the species richness of ants progressively declined with increasing elevation in subtropical Queensland, Australia. Previous results showed that temperature was a primary abiotic factor in diversity in response to spatial change (elevation or latitude). Binkenstein et al. [12] reported that the richness of predatory arthropods was lower at a higher elevation and pointed out this phenomenon was mainly related to the lower mean temperature. Parallel to the findings of a latitudinal study, Finch et al. [63] also found spider richness was greater in warm regions than in cold conditions.

At the local scale, vegetation (type or structure), as a primary biotic driver of diversity, often results in variations in spider communities [64,65]. This was supported by our results of regression analysis. The dominant trees and the structure of vegetation were altered with elevation changing and thus significantly decreased the richness of canopy spiders (Table 1, Figure 5B). At 800 m, the broadleaf trees (e.g., *T. amurensis* and *P. cathayana*) take great proportions, whereas the areas are mainly covered by conifer forests at 1400 m and 1700 m. We assume the vegetation configuration would result in lower vegetation complexity and heterogeneity at higher elevations and thus decrease species richness. Previous results demonstrated that there was a positive relationship between species richness and vegetation complexity [66,67]. Furthermore, the decrease in richness of canopy spiders with elevation was more dependent on tree height (Figure 5D) because higher trees generally have more complex canopy structures and larger canopy volumes. This corresponded with those of [68] in that forest height likely enhanced canopy arthropod diversity.

Notably, there was a sharp decline in richness between 1100 m and 1400 m in our results, maybe due to the distinguished changes in vegetation properties (Table 1). As shown, almost all vegetation variables had significant differences between 800 m and 1400 m and the characteristics were similar between 1400 m and 1700 m. We agree that there might be a transition zone between low elevations and high elevations [34,40,69] in CMNR, covered by mixed coniferous forests. Below and above the transition zone were mainly covered by mixed coniferous and broad-leaved forests and sub-alpine coniferous forests, respectively. Besides species richness, other diversity indices were also calculated and the responses to elevation change were different. The exponential Shannon diversity (^1^*D*) also substantially decreased with elevation increasing, whereas Pielou evenness index exhibited a slightly increasing trend (Figure 2E, *p* = 0.041). Inverse Simpson diversity (^2^*D*) and inverse Berger–Parker (^3^*D*) showed no significant differences among elevations. As known, ^1^*D* is more sensitive to rare species whereas ^2^*D* and ^3^*D* are more sensitive to dominant species. In total, ^0^*D* and ^1^*D* both showed decreasing patterns with elevation increasing. Species evenness is often regarded as a useful proxy for understanding the community structure and composition. In our study, the community composition significantly changed with elevation changing (see below). This indicates that it is necessary to pay attention to community evenness and structure when considering changes in community diversity.

### 4.2. Changes in Abundance and Composition of Canopy Spiders

Contrasting our first hypothesis, the total abundance of canopy spiders exhibited a hump-shaped pattern along an elevation gradient in our study. This was mainly because of a large increase in the abundance of Thomisids at 1100 m (Figure 3A,B). Two dominant Thomisids (*X. emertoni* and *L. silvanus*) took great proportions at 1100 m. The pattern was not in agreement with previous results. Otto and Svensson [70] showed a decreased elevation pattern on the abundance of ground spiders while Chatzaki et al. [71] found no significant effect on the abundance (activity) of Gnaphosids along five elevation zones (0–2400 m) in Crete. Similar to changes in diversity, changes in abundance might also be related to habitat characteristics (or vegetation). Moreover, the relation between abundance and vegetation was complex, not a simple linear pattern (in our study, *p* = 0.124).

β-diversity, as a measurement for community composition, is closely related to ecosystem function [72]. It has been well documented to generally decline with increasing elevation [73]. The decrease in β-diversity might be related to the increase in individuals of the same species or the decrease in individuals of different species in the community. Our results partly supported this point and the lowest value of β-diversity was obtained at 1100 m probably due to a large increase in individuals of Thomisids mentioned above (Figure 4A and Appendix A). The ordination map demonstrated that the variance in composition at lower elevations (between 800 m and 1100 m) was greater than that at higher elevations (between 1400 m and 1700 m) (Figure 4B). Similar results have been reported by other researchers [34,48], suggesting that habitats at higher elevations are more similar for animals. Furthermore, this phenomenon can also be supported by the endemic species at every elevation. Levels of spider endemicity mirrored the vegetation complexity and heterogeneity in the region [12]. Of the total number of species (81 species), 14 species (17.28%) were endemic to 800 m, 5 (6.17%) to 1100 m, 1 (1.23%) to 1400 m, and 2 (2.46%) to 1700 m (Figure 7). There were 19 shared species across the four elevations, accounting for 23.46% of the total number of species. Most of them had a strong capacity for ballooning, such as Linyphiids (6 species), Theridiids (4 species), and Araneids (3 species) [74,75]. There were 23 shared species between 800 m and 1100 m and 6 between 1400 m and 1700 m. There were three and five species mutually distributed at 800 m, 1100 m, 1400 m, 1100 m, 1400 m, and 1700 m, respectively. No shared species clearly existed between other combinations of elevations. Accordingly, the changes in spider composition were greater in low-elevation areas than in high-elevation areas. The results indicate that the β-diversity of canopy spiders is positively related to environmental heterogeneity.

The responses on the relative abundance of canopy spiders to elevation changes were different at the three taxonomic levels, which were in line with our second hypothesis. Taking family response as an example, the relative abundance of Theridiidae and Linyphiidae was significantly higher at 1400 m while Clubionidae demonstrated no significant differences with elevation change in the study. This was not in agreement with the findings of Russell-Smith and Stork in Indonesia [55], which showed that the proportion of Theridiidae was almost similar and that of Clubionidae was increasing with elevation. Sørensen [43] demonstrated that the importance of Linyphiidae increased with the elevation increasing. We assume that the different responses reflect preference and adaption to certain habitat characteristics. For instance, Mcnett and Rypstra [66] discussed the habitat selection of *Argiope trifasciata* (Araneidae, a large orb-weaving spider) and highlighted an important role in vegetation complexity. Koponen [76] found that Lycosidae was abundant in open habitats while Linyphiidae preferred forested habitats. The statement was in line with the results of our study that Linyphiidae ranked third in total abundance, yet Lycosidae was found only once. Some authors also attempted to explore what factors would be responsible for this kind of selection of habitats. Wise [77] pointed out that spider distribution was dependent on wind, moisture, and temperature. Entling et al. [78] compared spider communities from 70 habitat types in central Europe using correspondence analysis and characterized the distribution of spider species along two environmental gradients (shading and moisture). Especially, the preference can be obviously demonstrated by the relative abundance of the guild (Figure 3C, *p* < 0.01).

We found that at the species or family level, vegetation variables almost had significant effects on community composition, especially TC and HC being of greater importance at the species and family level, respectively (Figure 6A,B). This partly matched the findings, which highlighted the importance of vegetation cover [65,79]. It is noteworthy that the composition at the guild level showed no significance in the study, which supported the hypothesis that composition in the guild was more stable than taxonomic composition [80]. Additionally, this result indicated that the influence of elevation and vegetation factors weakened, in consistency with our third hypothesis. We suppose when considering community diversity or composition at different taxonomic levels, that the impact of factors would be different or completely changed. This was partly in line with the previous result. For example, Peters et al. [11] sampled 25 major plant and animal taxa on Mt. Kilimanjaro and found temperature would be the main predictor of species richness when scaling up diversity from single taxa to community level.

In fact, elevation change involves a compound influence resulting from many variables. In addition to vegetation factors, there are still many other important factors that might be responsible for the variation in canopy spider communities along elevation gradients, such as spatial (area and mid-domain-effect) and food resources at the local scale or climatic factors at large scale. Similar work has been conducted on ants or moths [9,81], yet there is little work involving these aspects on canopy spiders. Of course, the influence caused by human activities should also be considered [82]. No single factor appears adequate to account for the patterns. In combination with as many factors as possible, it would provide a comprehensive understanding of the effect of elevation on canopy spider communities.

## 5. Conclusions

We evaluated the changes in the diversity and composition of canopy spider communities with elevation changes in a typical temperate forest. Species richness decreased and evenness slightly increased, indicating the importance of community structure. The responses on the composition of canopy spiders to elevation changes were altered at different taxonomic levels (species, family, and guild). Our results support the widely held view that vegetation affects spider communities. Furthermore, the degree of influence of vegetation on composition would decrease with the taxonomic level rising. For our results obtained from a single sampling area, we admit that more work-related improvements are necessary in the future for generality and extension.

## Figures and Tables

**Figure 1 insects-15-00154-f001:**
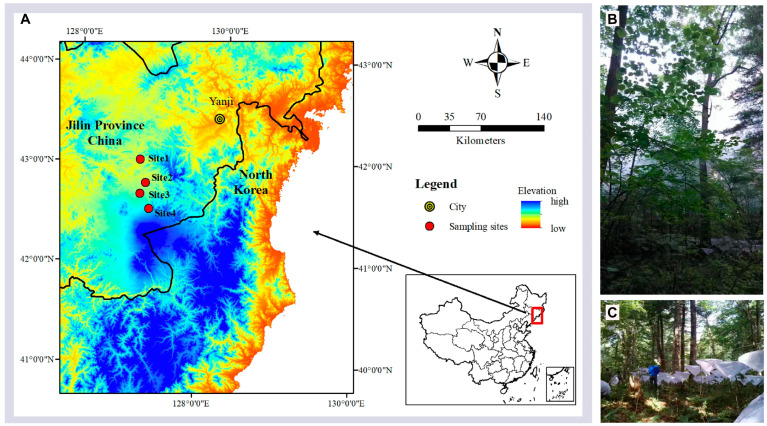
Sampling sites on Changbai Mountain, China. (**A**) Location. (**B**,**C**) Fogging process and funnel-like trays. Elevation: Site 1 = 800 m, Site 2 = 1100 m, Site 3 = 1400 m, and Site 4 = 1700 m.

**Figure 2 insects-15-00154-f002:**
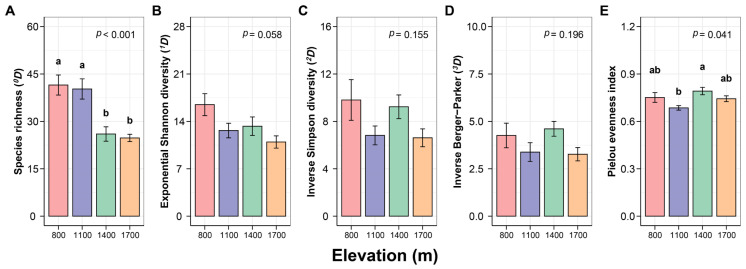
Diversity of canopy spiders to elevation changes on Changbai Mountain, China. (**A**) Species richness (^0^*D*). (**B**) Exponential Shannon diversity (^1^*D*). (**C**) Inverse Simpson diversity (^2^*D*). (**D**) Inverse Berger-Parker (^3^*D*). (**E**) Pielou evenness index. Error bar means standard error (S.E.). The number of replicates was 4 (*n* = 4). Lowercase letters indicate significant difference for multiple comparisons using LSD.

**Figure 3 insects-15-00154-f003:**
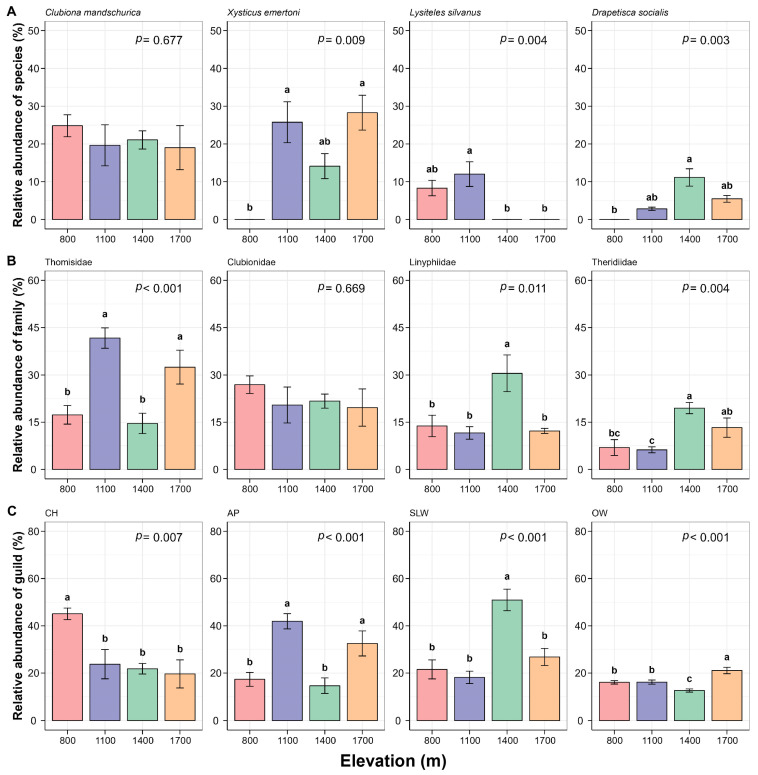
Relative abundance of species ((**A**), absolute abundance greater than 4%), family ((**B**), top four families), and guild (**C**) of canopy spiders to elevation changes on Changbai Mountain, China. Error bar means standard error (S.E.). The number of replicates was 4 (*n* = 4). Lowercase letters indicate significant differences for multiple comparisons using LSD. The tests of species in Figure 3A were examined using the Kruscal–Wallis test followed by the DUNN test for multiple comparisons.

**Figure 4 insects-15-00154-f004:**
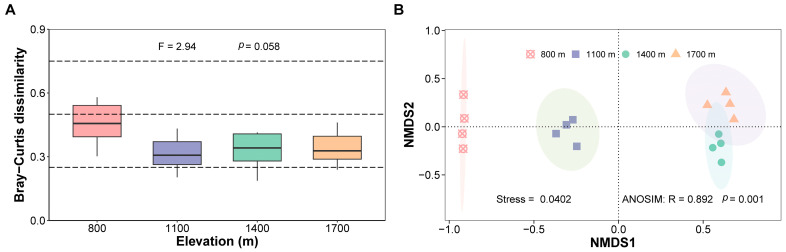
Variation in the community composition of canopy spiders among elevations on Changbai Mountain, China. (**A**) Bray–Curtis dissimilarity within the four elevations. (**B**) NMDS for canopy spider communities based on Bray–Curtis dissimilarity. The 95% confidence ellipses around group centroids. The number of replicates was 6 (*n* = 6) in Figure 4A. The dashed line means three levels of dissimilarity, with the Bray–Curtis distance being 0.25, 0.5, and 0.75 in Figure 4A.

**Figure 5 insects-15-00154-f005:**
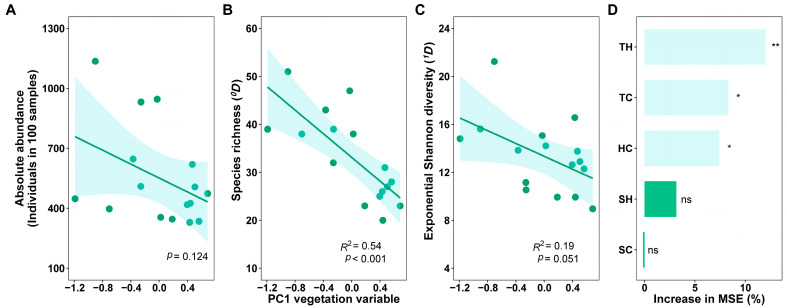
Responses of absolute abundance (**A**), species richness (**B**), and exponential Shannon diversity (**C**) of canopy spiders to vegetation factors and their relative importance on species richness (**D**) on Changbai Mountain, China. Light blue bars represent significant levels at *p* < 0.05 and green bars represent significant levels at *p* > 0.05 in Figure 5D. **, *p* < 0.01; *, *p* < 0.05; ns, no significance.

**Figure 6 insects-15-00154-f006:**
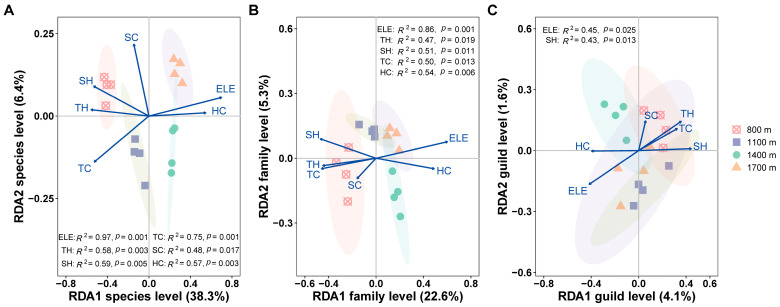
Habitat factors influencing the community composition of canopy spiders based on RDA on Changbai Mountain, China. (**A**) Species level. (**B**) Family level. (**C**) Guild level. Abbreviations: ELE = elevation, TH = tree height, SH = shrub height, TC = tree coverage, SC = shrub coverage, HC = herb coverage. The 95% confidence ellipses around group centroids. Significant influencing factors derived from permutational tests are shown (*p* < 0.05).

**Figure 7 insects-15-00154-f007:**
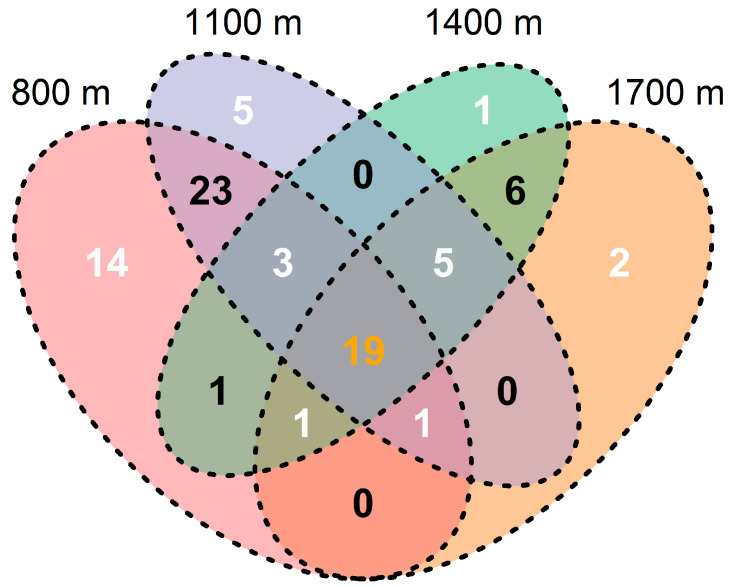
Venn diagram based on the composition of canopy spiders among elevations on Changbai Mountain, China. The different colors represent different elevations, and the different numbers indicate the shared and non-shared species among elevations.

**Table 1 insects-15-00154-t001:** Environmental variables of sampling plots at four elevations on Changbai Mountain, China (Mean ± S.E.).

Variable	Site 1	Site 2	Site 3	Site 4	*p*-Value
elevation (m)	800	1100	1400	1700	/
tree height (TH) (m)	26.3 ± 2.4 ^a^	23.0 ± 0.4 ^ab^	20.5 ± 0.5 ^ab^	19.5 ± 0.5 ^b^	0.021
shrub height (SH) (m)	5.4 ± 1.3 ^a^	3.3 ± 0.7 ^ab^	0.9 ± 0.1 ^c^	2.0 ± 0.5 ^bc^	0.002
tree coverage (TC) (%)	80.0 ± 2.9 ^a^	81.3 ± 1.5 ^a^	74.5 ± 0.5 ^b^	68.3 ± 1.2 ^c^	<0.001
shrub coverage (SC) (%)	57.5 ± 4.3 ^a^	22.5 ± 6.0 ^c^	35.0 ± 8.7 ^bc^	47.5 ± 7.5 ^ab^	0.018
herb coverage (HC) (%)	52.5 ± 10.5 ^b^	60.0 ± 4.1 ^b^	87.5 ± 2.5 ^a^	82.5 ± 4.8 ^a^	0.005
dominant tree species	*Pinus koraiensis*, *Tilia amurensis* and *Populus cathayana*	*P. koraiensis*, *T. amurensis* and *Abies nephrolepis*	*A. nephrolepis*, *Betula ermanii*, *Picea jezoensis* and *Larix olgensis*	*A. nephrolepis* and *L. olgensis*	/

Lowercase letters indicate significant difference for multiple comparisons using LSD. *p*-value of TH was obtained using Kruscal–Wallis test among the four elevations followed by the DUNN test for multiple comparisons.

## Data Availability

The original contributions presented in the study are included in the article/Appendix A; further inquiries can be directed to the corresponding author.

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
