# Peer review of "Vegetation Affects the Responses of Canopy Spider Communities to Elevation Gradients on Changbai Mountain, China"

_insects, 2024, doi:10.3390/insects15030154_

Round 1

Reviewer 1 Report

Comments and Suggestions for Authors

I read with interest your manuscript. However, I think that this manuscript at its current format is not acceptable for publication. It needs some additional analyses

There is a major problem that must be solved related with the outdated use of diversity Indices Shannon and Simpson in its original format as response variables. The use of the raw formulas of Shannon and Simpson is now considered inadequate since they are influenced by sampling bias.

Alternatively, Hills number allow a complete characterization of the diversity of a community. The authors should calculate species richness (Hill number N0), exponential Shannon diversity (Hill number N1), and inverse Simpson diversity (Hill number N2) and inverse Berger-Parker (Hill number N3). Hill numbers are order-sensitive, meaning that they can capture the influence of different species in a community. Depending on the order (q) chosen, you can emphasize the importance of dominant or rare species in the diversity calculation.

The four indices are knowing to give an emphasis respectively on total species (species richness), rare species (Shannon diversity), dominant species (Simpson diversity) and the top dominant species (Berger-Parker).
> The Hill numbers use a diversity profile organized in four orders (q) as follows: i) species richness (S) (q = 0), taken as a count of the number of species in a particular site; ii) the exponential Shannon-Wiener (exp H´) (q = 1); iii) the inverse of Simpson´s concentration index (1/D) (q = 2) and inverse Berger-Parker (1/d) (q = 3).
The Hill numbers are very rich in information, since they combine information on species richness, species rarity (species relative abundances) and species dominance.

Hill, M.O. Diversity and Evenness: A Unifying Notation and Its Consequences. Ecology 1973, 54, 427–432, doi:https://doi.org/10.2307/1934352

Jost, L. Partitioning diversity into independent alpha and beta components. Ecology 2007, 88, 2427–2439, doi:https://doi.org/10.1890/06-1736.1

Chao, A., Chiu, C. H., & Jost, L. (2014). Unifying species diversity, phylogenetic diversity, functional diversity, and related similarity and differentiation measures through Hill numbers. Annual review of ecology, evolution, and systematics45, 297-324.

My best regards,

Author Response

Response Letter

Dear Editor,

We appreciate the opportunity to revise our manuscript and thank all the reviewers for the constructive assessments. All suggestions were addressed in our revision and the response letter, and each comment has received a specific response (highlighted), marked by line numbers. We believe the manuscript has benefited greatly from the editor's and reviewers' concerns and suggestions. We are very grateful for your time, effort, and consideration of our work. Please let us know if you have any questions regarding the revision of our manuscript and/or responses to the reviewer comments.

Besides, we have revised the whole manuscript carefully and we further asked professional language editing services to refine the language, though the Reviewer 2 said “The use of the English language is appropriate and comprehensible”.

Guo Zheng and co-authors

Reviewer 1

I read with interest your manuscript. However, I think that this manuscript at its current format is not acceptable for publication. It needs some additional analyses

There is a major problem that must be solved related with the outdated use of diversity Indices Shannon and Simpson in its original format as response variables. The use of the raw formulas of Shannon and Simpson is now considered inadequate since they are influenced by sampling bias.

Alternatively, Hills number allow a complete characterization of the diversity of a community. The authors should calculate species richness (Hill number N0), exponential Shannon diversity (Hill number N1), and inverse Simpson diversity (Hill number N2) and inverse Berger-Parker (Hill number N3). Hill numbers are order-sensitive, meaning that they can capture the influence of different species in a community. Depending on the order (q) chosen, you can emphasize the importance of dominant or rare species in the diversity calculation.

The four indices are knowing to give an emphasis respectively on total species (species richness), rare species (Shannon diversity), dominant species (Simpson diversity) and the top dominant species (Berger-Parker).

The Hill numbers use a diversity profile organized in four orders (q) as follows: i) species richness (S) (q = 0), taken as a count of the number of species in a particular site; ii) the exponential Shannon-Wiener (exp H´) (q = 1); iii) the inverse of Simpson´s concentration index (1/D) (q = 2) and inverse Berger-Parker (1/d) (q = 3).

The Hill numbers are very rich in information, since they combine information on species richness, species rarity (species relative abundances) and species dominance.

Hill, M.O. Diversity and Evenness: A Unifying Notation and Its Consequences. Ecology 1973, 54, 427–432, doi:https://doi.org/10.2307/1934352

Jost, L. Partitioning diversity into independent alpha and beta components. Ecology 2007, 88, 2427–2439, doi:https://doi.org/10.1890/06-1736.1

Chao, A., Chiu, C. H., & Jost, L. (2014). Unifying species diversity, phylogenetic diversity, functional diversity, and related similarity and differentiation measures through Hill numbers. Annual review of ecology, evolution, and systematics, 45, 297-324.

Thanks for the reviewer's suggestions and helpful information. We have changed these diversity indices to Hill numbers according to the suggestions (lines 143-146, lines 201-209, lines 263-266, Figure 2 and Figure 5).

Reviewer 2 Report

Comments and Suggestions for Authors

The research exhibits a strong framework and highlight the quantity of work accomplished, particularly evident in the substantial number of specimens collected. However, I have several recommendations that I believe are important for enhancing the overall quality of the manuscript.

While the title is a personal choice, a more concise formulation could potentially enhance its impact. The current title appears excessively lengthy for a first impression.

To improve the visual appeal and comprehensiveness of the paper, it is advisable to include images or photographs illustrating the four sample sites. The addition of visuals would significantly contribute to the aesthetic appeal of the work.

Figure 1 requires revision to incorporate the photographs, and the caption line to the figure also needs improvements and completion. The lack of descriptions for the photographs in the caption diminishes clarity and impairs the comprehension of the figure.

In the discussion section, it is advisable to address the potential influence of the chosen sample month. While August was selected for sampling due to the heightened spider activity, it is important recognize that certain species may exhibit activity peak during other seasons. Given that sampling in other seasons is not feasible, it would be beneficial that the manuscript include some sentences in the discussion for exhibit this limitation and discuss its implications.

A critical aspect lacking in the methods section is detailed information regarding the fogging technique. It is imperative to provide additional details about the fogging process, include the specific product utilized.    

Comments on the Quality of English Language

The use of the English language is appropriate and comprehensible; only some errors have been identified that could be reviewed (e.g., in line 269, could be: for example).

Author Response

Response Letter

Dear Editor,

We appreciate the opportunity to revise our manuscript and thank all the reviewers for the constructive assessments. All suggestions were addressed in our revision and the response letter, and each comment has received a specific response (highlighted), marked by line numbers. We believe the manuscript has benefited greatly from the editor's and reviewers' concerns and suggestions. We are very grateful for your time, effort, and consideration of our work. Please let us know if you have any questions regarding the revision of our manuscript and/or responses to the reviewer comments.

Besides, we have revised the whole manuscript carefully and we further asked professional language editing services to refine the language, though the Reviewer 2 said “The use of the English language is appropriate and comprehensible”.

Guo Zheng and co-authors

Reviewer 2

The research exhibits a strong framework and highlight the quantity of work accomplished, particularly evident in the substantial number of specimens collected. However, I have several recommendations that I believe are important for enhancing the overall quality of the manuscript.

1) While the title is a personal choice, a more concise formulation could potentially enhance its impact. The current title appears excessively lengthy for a first impression.

Thanks for the reviewer's suggestion, the title has been changed by “Vegetation affects the responses of canopy spider communities to elevation gradients on Changbai Mountain, China” (lines 2-4).

2) To improve the visual appeal and comprehensiveness of the paper, it is advisable to include images or photographs illustrating the four sample sites. The addition of visuals would significantly contribute to the aesthetic appeal of the work.

We believe this suggestion is very valuable. Unfortunately, we are very sorry to say that we cannot find high-quality photos that can be published for all four sample sites.

3) Figure 1 requires revision to incorporate the photographs, and the caption line to the figure also needs improvements and completion. The lack of descriptions for the photographs in the caption diminishes clarity and impairs the comprehension of the figure.

Done. The caption of Figure 1 was changed as follows:

Figure 1. Sampling sites on Changbai Mountain, China. (A) location. (B/C) fogging process and funnel-like trays. Elevation: Site 1= 800 m, Site 2= 1100 m, Site 3= 1400 m, Site 4= 1700 m (lines 104-105).

4) In the discussion section, it is advisable to address the potential influence of the chosen sample month. While August was selected for sampling due to the heightened spider activity, it is important recognize that certain species may exhibit activity peak during other seasons. Given that sampling in other seasons is not feasible, it would be beneficial that the manuscript include some sentences in the discussion for exhibit this limitation and discuss its implications.

Thanks for the reviewer's suggestion, we have added some information in discussion according to the suggestion (lines 298-300).

5) A critical aspect lacking in the methods section is detailed information regarding the fogging technique. It is imperative to provide additional details about the fogging process, include the specific product utilized.

Done. We added more detailed descriptions about the fogging machine (lines 122-123), pesticide and the fogging process (lines 129-130).

6) The use of the English language is appropriate and comprehensible; only some errors have been identified that could be reviewed (e.g., in line 269, could be: for example).

Done (line 306).

Round 2

Reviewer 1 Report

Comments and Suggestions for Authors

The authors performed all needed changes. Now the manuscript can be accepted.